# Inhibitory Effects of *Jiuzao* Polysaccharides on Alcoholic Fatty Liver Formation in Zebrafish Larvae and Their Regulatory Impact on Intestinal Microbiota

**DOI:** 10.3390/foods13020276

**Published:** 2024-01-16

**Authors:** Qing Li, Liling Wu, Guangnan Wang, Fuping Zheng, Jinyuan Sun, Yuhang Zhang, Zexia Li, Lianghao Li, Baoguo Sun

**Affiliations:** 1Key Laboratory of Geriatric Nutrition and Health, Ministry of Education, Beijing Technology and Business University, Beijing 100048, Chinasunjinyuan@btbu.edu.cn (J.S.); 18895706570@163.com (L.L.); sunbg@btbu.edu.cn (B.S.); 2Key Laboratory of Brewing Molecular Engineering of China Light Industry, Beijing Technology and Business University, Beijing 100048, China; 3Beijing Laboratory of Food Quality and Safety, Beijing Technology and Business University, Beijing 100048, China; 4Hebei Hengshui Laobaigan Liquor Co., Ltd., Hengshui 053009, Chinahslzx999@163.com (Z.L.)

**Keywords:** *Jiuzao* polysaccharide, zebrafish, alcoholic fatty liver, gene expression, microbiome

## Abstract

The liver is critical in alcohol metabolism, and excessive consumption heightens the risk of hepatic damage, potentially escalating to hepatitis and cirrhosis. *Jiuzao*, a by-product of *Baijiu* production, contains a rich concentration of naturally active polysaccharides known for their antioxidative properties. This study investigated the influence of *Laowuzeng Jiuzao* polysaccharide (LJP) on the development of ethanol-induced alcoholic fatty liver. Zebrafish larvae served as the model organisms for examining the LJPs hepatic impact via liver phenotypic and biochemical assays. Additionally, this study evaluated the LJPs effects on gene expression associated with alcoholic fatty liver and the composition of the intestinal microbiota through transcriptomic and 16 S rRNA gene sequencing analyses, respectively. Our findings revealed that LJP markedly mitigated morphological liver damage and reduced oxidative stress and lipid peroxidation in larvae. Transcriptome data indicated that LJP ameliorated hepatic fat accumulation and liver injury by enhancing gene expression involved in alcohol and lipid metabolism. Furthermore, LJP modulated the development of alcoholic fatty liver by altering the prevalence of intestinal Actinobacteriota and Firmicutes, specifically augmenting *Acinetobacter* while diminishing *Chryseobacterium* levels. Ultimately, LJP mitigated alcohol-induced hepatic injury by modulating gene expression related to ethanol metabolism, lipid metabolism, and inflammation and by orchestrating alterations in the intestinal microbiota.

## 1. Introduction

The traditional custom of drinking is prevalent across various cultures; however, its excessive use poses significant health risks. The liver, the primary site for alcohol metabolism, is particularly vulnerable to oxidative stress caused by excessive alcohol intake. This oxidative stress can lead to the accumulation of fat within the liver cells, resulting in alcoholic fatty liver disease (AFLD) and increasing the risk of hepatic injuries that may progress to hepatitis and cirrhosis [1,2]. Consequently, the impact of alcohol on liver function has become a focal point of scientific inquiry [3]. *Jiuzao* is a solid mixture of fermented grain residue after distilling, and it is the most dominant by-product of the *Baijiu* industry, containing rich crude protein, dietary fiber, trace elements, and vitamins. Currently, *Jiuzao* is underutilized, often relegated to animal feed or organic fertilizer, leading to resource wastage and environmental concerns.

Research has revealed that certain natural polysaccharides can bolster the production of immune cells, enhance resistance to pathogens, and offer a protective or therapeutic effect against alcoholic liver damage [4]. To date, over 95 natural active polysaccharides have been identified for liver protection [5]. Polysaccharides from *Ganoderma lingzhi* have shown efficacy in preventing acute alcoholic liver injury in mice by modulating genes linked to glutathione metabolism, cytochrome P450, and retinol metabolism [6]. Zhang et al. reported that *Lycium barbarum* polysaccharides not only promote the growth of Nile tilapia but also safeguard the fish’s spleen and liver [7]. These active polysaccharides prevent lipid peroxidation, thus preserving cell membrane structure and function [8]. The addition of fermented wheat bran polysaccharide in feed could increase the relative abundance of Firmicutes, thereby regulating the intestinal microbiota and then promoting the growth of zebrafish [9]. Furthermore, *Jiuzao* polysaccharides have demonstrated antioxidative properties in zebrafish, acting as free radical scavengers and indicating potential as hepatoprotective agents [10].

The zebrafish (*Danio rerio*) model for alcoholic liver disease facilitates the rapid identification and analysis of genes implicated in hepatic steatosis [5]. By four days post-fertilization (dpf), zebrafish larvae possess a sufficiently mature liver capable of synthesizing essential enzymes for alcohol metabolism, thus establishing it as an excellent model for studying alcoholic liver disease and for conducting related pharmaceutical screenings [11,12]. A prior investigation demonstrated that zebrafish larvae immersed in a 2% ethanol culture medium for four dpf achieved an intracellular ethanol concentration of 80 mM after 32 h. This condition led to the upregulation of cpy2e1, sod, and bip genes in the larvae’s livers, indicating ethanol metabolism and consequent oxidative stress [13]. Metabolic profiling using proton nuclear magnetic resonance spectroscopy (NMR) and gas chromatography-mass spectrometry uncovered metabolic disparities between the alcohol-treated and control zebrafish, confirming that alcohol induces oxidative stress and that metabolites in fatty liver tissues correlate with chronic alcohol consumption [14]. Transcriptome sequencing is increasingly used to observe changes in mRNA expression post-treatment, playing a significant role in the understanding of pathogenesis, disease marker screening, and the diagnosis and treatment of liver injury, among other diseases [15].

The intestinal microbiota is crucial in the progression of liver injury, with the intricate relationship between gut metabolism and liver function becoming more apparent [16,17]. This microbiota partakes in alcohol metabolism, transforming ethanol into toxic acetaldehyde, thus disrupting the integrity of the intestinal mucosa and increasing intestinal permeability, which is often called intestinal leakage [18]. Moreover, excessive ethanol consumption disrupts the balance of intestinal flora, promoting the proliferation of Gram-negative bacteria, the production of endotoxin-induced tumor necrosis factor (TNF-α), the activation of the inducible nitric oxide synthase system (iNOS), and intensifying the intestinal inflammatory response. The resultant surge in NO synthesis fosters oxidative stress in the liver, exacerbating hepatic damage [19].

While previous studies have recognized the antioxidant capacity of natural polysaccharides in preventing acute alcoholic liver injury in mice, reports on the effects of *Jiuzao* polysaccharides, particularly on lipid metabolism changes in the liver and intestinal bacterial regulation, are scarce. Accordingly, this study employed transcriptomics analysis to investigate the influence of *Jiuzao* polysaccharide on differentially expressed genes (DEGs) associated with ethanol-induced liver injury in zebrafish larvae. Additionally, the changes in intestinal bacteria in zebrafish larvae were observed by 16 S rRNA gene sequencing technology in order to explore the role of *Jiuzao* polysaccharide in the occurrence and development of alcoholic fatty liver. It is well known that long-term excessive drinking can cause serious damage to the liver. Therefore, in order to prepare healthy alcoholic drinks and realize high-value reuse of *Jiuzao*, it is considered necessary to extract active polysaccharides from *Jiuzao* and add them back to *Baijiu*. In this paper, polysaccharides were extracted from *Jiuzao,* and their biological activities were studied, which provided ideas for preparing healthy alcoholic beverages and improving the reuse of *Jiuzao’s* high-value.

## 2. Materials and Methods

### 2.1. Materials and Chemicals

The *Jiuzao* sample was obtained from a workshop for *Laowuzeng Baijiu* at Hebei Hengshui Laobaigan Liquor Industry Co., Ltd., Hengshui, China, in 2021 [20,21]. A 0.5% saturated solution of Oil Red O, 0.01 M PBS (powder, pH 7.2–7.4), hematoxylin and eosin (H&E) staining kit, Nile red, DAPI solution, Triton X-100, and kits for superoxide dismutase (SOD) activity, catalase (CAT) activity, glutathione (GSH) content, and malondialdehyde (MDA) content were obtained from Beijing Solarbio Science and Technology Co., Ltd. (Beijing, China). The embryonic medium was sourced from Nanjing Eze-Rinka Biotechnology Co., Ltd., Nanjing, China. All other chemicals used were of analytical grade and procured from local suppliers in China.

### 2.2. Preparation of Polysaccharides

One thousand grams of dried *Jiuzao* powder were extracted three times with cold water (1:20 *w*/*v*), each for 1.2 h. The extracts were concentrated, and ethanol was added to achieve an 80% ethanol concentration in the solution, which was then left to precipitate overnight. The mixture was centrifuged at 5000 rpm for 10 min to collect the precipitate. The Sevag method was applied for deproteinization, yielding crude polysaccharides. Following Li’s method [10], the polysaccharides were further purified using DEAE-Sepharose Fast Flow and Sephadex G-50 columns, resulting in pure *Laowuzeng Jiuzao* polysaccharides (LJP). Through molecular weight determination, monosaccharide composition analysis, methylation, and NMR studies, LJP was characterized as a polysaccharide predominantly composed of mannose with an average molecular weight of 32,402 g/mol.

### 2.3. Animals and Experimental Design

This study’s protocol received approval from the Ethics Committee for Laboratory Animal Welfare and Animal Experimentation at Beijing Technology and Business University (Lot No. AW110099102-3-4). Wild-type AB strain zebrafish larvae at 3 days post-fertilization (3 dpf) and liver-specific transgenic expression (Tg [*apo14:EGFP*]) at 3 dpf were sourced from Eze-Rinka Biotechnology Co., Ltd. The larvae were cultured under a 14-h light/10-h dark cycle at 28 °C in aerated embryonic medium (26 ± 1 °C), with a composition of 5 mM NaCl, 0.17 mM KCl, 0.4 mM CaCl_2_, and 0.16 mM MgSO_4_ at a pH of 6.7, in accordance with ‘The Zebrafish Book: A Guide for the Laboratory Use of Zebrafish (*Danio rerio*)’ published by the University of Oregon Press.

The healthy larvae at 6 days post-fertilization (dpf) were randomly allocated to each well of a 6-well plate, with 50 larvae per well. The experimental setup included four groups: control (C), model (M), positive drug groups (A_1, A_2, A_3), and LJP groups (LJP_1, LJP_2, LJP_3) (Figure 1). The C group was maintained in zebrafish embryonic medium; the M group was cultured in embryonic medium with 350 mM ethanol; the positive drug groups received atomolan at 0.012, 0.025, and 0.05 mg/mL, respectively, in embryonic medium with 350 mM ethanol; and the LJP groups received LJP at 0.05, 0.1, and 0.2 mg/mL in embryonic medium with 350 mM ethanol. Each group had three replicates with 50 healthy larvae. After 32 h of treatment, observations were made for larval deformities and mortality, leading to further experimental procedures.

### 2.4. Oil Red O Staining

The wild-type AB strain zebrafish larvae were anesthetized with a 0.02% tricaine solution, fixed in 4% paraformaldehyde (PFA) at 4 °C overnight, and subsequently rinsed three times with phosphate-buffered saline (PBS). Then, the larvae were dehydrated in a graded series of propylene glycol solutions (20%, 40%, 80%, and 100%) for 15 min each. The larvae were stained with a 0.5% Oil Red O solution and left at room temperature, shielded from light, for 12 h. Following staining, the larvae were briefly dipped in 100% and 80% propylene glycol to remove excess dye and then cleared with PBS. A Nikon optical microscope (Tokyo, Japan) was used to examine hepatic morphology and lipid droplet accumulation. Three representative images were captured for analysis. Using ImageJ software (version 1.53), the staining intensity and liver size were quantified as grayscale values to evaluate the extent of hepatic steatosis.

### 2.5. Hematoxylin and Eosin Stainings

The procedure was conducted following the method described by Zhou et al. [22]. The wild-type AB strain zebrafish larvae were anesthetized with a 0.02% tricaine solution and fixed in 4% PFA at 4 °C overnight, embedded in paraffin, and sectioned at 3–5 μm thickness. The sections were dewaxed in xylene for 5–10 min and rehydrated through a decreasing ethanol series (100%, 90%, 80%, and 70%) for 2 min each, followed by rehydration in distilled water for 2 min. The sections were stained with hematoxylin for 8 min, rinsed under tap water for 10 min, and briefly washed in distilled water, followed by eosin staining for 1 min. They were then dehydrated in an ascending ethanol series and cleared in xylene for 5 min. Finally, the sections were mounted with neutral gum and observed microscopically.

### 2.6. Nile Red and DAPI Stainings

For liver-specific analysis, EGFP transgenic zebrafish larvae *Tg* [*apo14:EGFP*] from each group were used, with 10 larvae per group. The larvae were anesthetized with a 0.02% tricaine solution, fixed overnight with 4% PFA, and then washed three times with PBS. The next day, the larvae were transferred to 96-well plates and permeabilized with a citric acid solution containing 0.1% Triton X-100. After three additional PBS washes, they were stained with DAPI at room temperature in the dark for 20 min to label the nuclei. This was followed by three PBS washes before staining with a Nile red solution (0.5 μg/mL, prepared by diluting a 0.5 mg/mL acetone stock solution with 75% glycerol) for 30 min in the dark at room temperature to stain the lipid droplets in the liver. Three larvae from each group were randomly selected for examination and imaging under a fluorescence microscope.

### 2.7. Enzymatic Assays in Ethanol-Induced Fatty Liver Disease in Zebrafish Larvae

To assess the impact of AFLD on enzyme activity in zebrafish larvae, commercial kits from Beijing Solarbio Science and Technology Co., Ltd., Beijing, China, were used following their guidelines. For each assessment, three sets of replicates with 50 larvae in each were prepared. The larvae were anesthetized with a 0.02% tricaine solution, homogenized in Tris-buffer under chilled conditions, and then centrifuged at 8000× *g* for 10 min at 4 °C. Through analyzing the supernatant, the activities of antioxidant enzymes, including superoxide dismutase (SOD) and catalase (CAT), and the contents of lipid peroxides, including glutathione (GSH) and malondialdehyde (MDA), were measured. Protein concentrations were determined using the bicinchoninic acid assay.

### 2.8. Tissue RNA Extraction, Library Construction, and Transcriptome Sequencing

Transcriptome analysis was performed on the C, M, A_3, and LJP_3 groups. Each set of three replicates contained 50 zebrafish larvae per replicate. After anesthesia with a 0.02% tricaine solution, the samples were washed three times with DEPC-treated water and placed in RNase-free 1.5 mL centrifuge tubes, 50 larvae per tube and three tubes per group. Total RNA was extracted using Trizol reagent (Invitrogen Corporation, Carlsbad, CA, USA). The RNA isolation, library preparation, and sequencing were conducted by Shanghai Majorbio Bio-Pharm Biotechnology Co., Ltd. (Shanghai, China). RNA integrity was assessed using an Agilent 2100 Bioanalyzer, and RNA concentration was determined with a Nanodrop ND-2000. Libraries were prepared with the TruSeq RNA Sample Preparation Kit and sequenced. Post-sequencing, reads were filtered using SeqPrep and Sickle for quality, then aligned to the reference zebrafish genome (GRCz11) with Tophat [23]. Gene expression levels were quantified as transcripts per million reads (TPM) by RSEM (http://deweylab.github.io/RSEM/ accessed on 1 October 2023) [24]. Differential expression analysis was performed using DESeq2, with genes showing over two-fold changes and *p* < 0.05, as well as an FDR < 0.05, considered differentially expressed. Gene ontology (GO) and Kyoto Encyclopedia of Genes and Genomes (KEGG) pathway enrichment analyses were carried out using the OmicShare tools (http://www.omicshare.com/tools/ accessed on 1 October 2023) [25], and protein-protein interaction networks were analyzed using Cytoscape and the STRING database (https://string-db.org/ accessed on 1 October 2023).

### 2.9. Real-Time Quantitative Polymerase Chain Reaction (RT-qPCR) Analysis 

The total RNA of the C, M, A_3, and LJP_3 groups was transcribed. Each set of three replicates contained 50 zebrafish larvae per replicate. The larvae were anesthetized with a 0.02% tricaine solution and immediately frozen in liquid nitrogen. The tissue RNA was reverse-transcribed into cDNA using HiScript Q RT SuperMix for qPCR (+gDNA wiper kit). Fluorescence quantification was performed using the cDNA as a template according to the instructions of the 2X ChamQ SYBR COLOR qPCR Master Mix kit to detect the expression level of the target gene. The ribosomal protein P0 (rpp0) gene was used as the internal reference gene, and the relative expression of the gene was calculated using the 2^−ΔΔCt^ method. The primer sequences are shown in Table 1.

### 2.10. 16 S rRNA Gene Sequencing of Intestinal Bacteria

The intestinal microbiota composition was analyzed using 16 S rRNA gene sequencing in the C, M, A_3, and LJP_3 groups. Each set contained three replicates with 50 larvae each. The larvae were anesthetized with a 0.02% tricaine solution and immediately frozen in liquid nitrogen. The total genomic DNA was isolated with a DNeasy PowerSoil Pro kit (MoBio Laboratories, Carlsbad, CA, USA) according to the manufacturer’s instructions [26]. DNA quality was verified by 1% agarose gel electrophoresis, and DNA concentration and purity were assessed using a NanoDrop 2000 spectrophotometer. The V3-V4 regions of the 16 S rRNA genes were amplified by PCR with primers 338F (5′-ACTCCTACGGGAGGCAGCAG-3′) and 806R (5′-GGACTACHVGGGTWTCTAAT-3′). The PCR products were mixed, recovered from a 2% agarose gel, and purified using the AxyPrep DNA Gel Extraction Kit (Axygen Biosciences, Union City, CA, USA). The products were quantified using a Quantus™ Fluorometer (Promega, Madison, WI, USA). Libraries were constructed with the NEXTFLEX Rapid DNA-Seq Kit and sequenced on the Illumina MiSeq PE300 platform.

Raw sequences were subjected to quality control using Fastp software (version 0.20.0) and assembled with Flash software (version 1.2.7). Sequence clustering into OTUs was performed with Uparse software (version 1.2.7), applying a 97% similarity threshold to remove chimeras. Taxonomic annotation of OTU representative sequences was based on the Silva 16 S rRNA gene database (v138), using the RDP classifier with a confidence threshold of 0.7. Significant differences in species across samples were analyzed with the Kruskal–Wallis H test. The OTU abundance table was standardized by PICRUSt to account for variations in the 16S gene copy numbers across species genomes. Functional annotation of OTUs was then conducted using their corresponding GreenGene IDs, yielding information on COG functions and the abundance of each function across samples.

### 2.11. Statistical Analysis

All experiments were performed in triplicate, with results expressed as the mean ± standard deviation. The data were analyzed using SPSS 19.0 statistical software. For three-sample comparisons, statistical significance was determined using one-way ANOVA, followed by Duncan’s post hoc test for multiple comparisons. Two-sample comparisons were conducted using an unpaired 2-tailed Student’s *t*-test. A *p*-value of 0.05 or lower was deemed indicative of statistical significance.

## 3. Results

### 3.1. Effect of LJP on Liver Visual Phenotype in AFLD Zebrafish Larvae

The Oil Red O staining results demonstrated (Figure 2A) that both the abdominal redness and liver enlargement in zebrafish larvae from the atomolan (A) and LJP groups were less pronounced than in the M group. Quantification of liver fat staining intensity, converted to grayscale values via ImageJ software (version 1.53), revealed that the fat staining intensity in the A group at 0.12, 0.025, and 0.05 mg/mL was reduced by 0.28, 0.45, and 0.54 times, respectively, compared to the M group. Similarly, in the LJP group, at concentrations of 0.05, 0.10, and 0.20 mg/mL, the intensity decreased by 0.27, 0.48, and 0.48-fold, respectively. The addition of atomolan and LJP mitigated liver steatosis (Figure 2C). Analysis of Figure 2A–C indicated that LJP successfully reduced liver fat accumulation in zebrafish larvae. As depicted in Figure 2D, ethanol exposure increased the cardiac rate of zebrafish larvae, while the A and LJP groups showed a gradual normalization of cardiac rate when compared to the M group (*p* < 0.01).

H&E-stained liver sections (Figure 2E) exhibited that hepatocytes in the C group maintained normal morphology with round nuclei and homogenous, abundant cytoplasm, arranged tightly and orderly. Conversely, larvae from the M group exhibited steatosis, with hepatocytes displaying extensive vacuolar degeneration, disorganized arrangements, and increased intercellular spaces. Larvae in the LJP group showed fewer lipid droplets than those in the M group, indicating that ethanol consumption disrupts hepatocyte integrity, enhances hepatic lipid droplet accumulation, and induces steatosis, while *Jiuzao* polysaccharides ameliorate ethanol-induced hepatic fat accumulation and reduce steatosis.

Figure 2F presents Nile red and DAPI staining results, demonstrating that larvae from the A and LJP groups exhibited less intense red, blue, and green fluorescence in a dose-dependent manner compared to the M group. This finding further indicates that LJP significantly reduces both liver and systemic fat accumulation in zebrafish larvae, thereby inhibiting steatosis.

### 3.2. Effect of LJP on Oxidative Stress and Lipid Peroxidation Indices in AFLD Zebrafish Larvae

Alcohol-induced oxidative stress and lipid peroxidation critically affect liver health. To evaluate the protective effects of LJP against oxidative stress, we measured the activities of SOD, CAT, and GSH, as well as levels of MDA. As indicated in Figure 3, SOD and CAT activities were lower in the M group than in the C group, signifying ethanol’s substantial damage to zebrafish larvae. However, SOD and CAT levels increased in the A and LJP groups across all doses compared to the M group. The most notable increase in CAT activity was observed following high-dose LJP treatment, suggesting that *Jiuzao* polysaccharides can mitigate oxidative stress in zebrafish larvae. Furthermore, while the GSH content was significantly reduced in the M group (*p* < 0.01), it increased in the LJP group in a dose-dependent manner. MDA content, a marker of lipid peroxidation, was significantly lower in the 0.05, 0.1, and 0.2 mg/mL LJP-treated groups compared to the M group (*p* < 0.01), indicating an enhancement in antioxidant capacity due to *Jiuzao* polysaccharide treatment.

### 3.3. Effect of LJP on the Expression of Liver Lipid-Related Genes in AFLD Zebrafish Larvae

#### 3.3.1. Differentially Expressed Gene Profiles

The volcano plots (Figure 4A–C) display the differential expression profiles among the groups (C vs. M, A vs. M, and LJP vs. M). After screening for differentially expressed genes (DEGs) with an FDR ≤ 0.05 and |log2 fold change (FC)| ≥ 1.5, the comparison between the C and M groups yielded 3168 DEGs from 27,127 genes, with 2257 up-regulated and 911 down-regulated (Figure 4A). The A group, when compared with the M group, showed 2938 DEGs from 27,361 genes, with 1889 up-regulated and 1049 down-regulated (Figure 4B). In the LJP group, compared to the M group, there were 2861 DEGs from 27,369 genes, with 1889 up-regulated and 972 down-regulated (Figure 4C). The GO and KEGG pathway enrichment analyses of DEGs are illustrated in histograms (G,H) and bubble plots (I,J), while an interaction map of the differentially expressed protein-protein network is depicted (K).

#### 3.3.2. Gene Annotation and Functional Analysis

The GO annotations of DEGs are shown in Figure 4D,E. The C vs. M and the LJP vs. M all showed the following results: In the category of Biological Process, most genes were annotated for cellular processes and biological regulation.

To delve into the impact of *Jiuzao* polysaccharides on the principal signaling pathways in AFLD in zebrafish larvae, we conducted KEGG enrichment analyses on DEGs from both the control versus model (C vs. M) and *Jiuzao* versus model (LJP vs. M) comparisons. As depicted in Figure 4F, the DEGs in zebrafish larvae exposed to ethanol (C vs. M groups) were predominantly associated with pathways like chemical carcinogenesis-reactive oxygen species, apoptosis, IL-17 signaling, hepatitis B, and non-alcoholic fatty liver disease. These findings imply that ethanol exposure alters multiple genes linked to lipid metabolism, oxidative stress, and inflammation in zebrafish larvae.

Peroxisome proliferator-activated receptors (PPARs) and sterol regulatory element-binding proteins (SREBPs) are key regulators affected by imbalances in fatty acid oxidation and synthesis, contributing to AFLD [27]. Figure 4F reveals that DEGs in the LJP and M groups are significantly involved in pathways such as fatty acid degradation, PPAR signaling, fat digestion and absorption, metabolism of xenobiotics by cytochrome P450, drug metabolism-cytochrome P450, steroid hormone biosynthesis, and fatty acid elongation. This suggests a potential role for *Jiuzao* polysaccharides in modulating lipid and ethanol metabolism in AFLD-affected zebrafish larvae.

#### 3.3.3. Protein-Protein Interaction (PPI) Network for Lipid Metabolism-Related Genes

A PPI network was established for DEGs associated with fatty acid degradation, steroid biosynthesis, and ethanol metabolism. The hub genes demonstrated robust connectivity. As shown in Figure 4H, 124 DEGs enriched in chemical carcinogenesis, reactive oxygen species, apoptosis, the IL-17 signaling pathway, hepatitis B, and non-alcoholic fatty liver disease pathways were selected from 3168 DEGs in the C vs. M groups. PPI network analysis confirmed that *jun*, *apaf1*, *cebpb*, *il1b*, *atp5fa1*, and others had strong biological interactions. As illustrated in Figure 4I, of the 2861 DEGs between the LJP and M groups, 162 were enriched in pathways such as fatty acid degradation, PPAR signaling, fat digestion and absorption, xenobiotic metabolism by cytochrome P450, drug metabolism by cytochrome P450, steroid biosynthesis, steroid hormone biosynthesis, and fatty acid elongation. The PPI network analysis confirmed strong biological interactions among *acox1*, *zgc:77938*, *cyp3a65*, *ugt1a1,* and *apoba*.

#### 3.3.4. qRT-PCR Validation Analysis of DEGs

Quantitative reverse transcription PCR (qRT-PCR) was used to validate the expression of the *jun*, *hsp90aa1.2*, *acox1*, *cyp3a66*, and *fabp2* genes (Figure 4J). Compared with the C group, the expressions of *jun* and *hsp90aa1.2* in group M were significantly up-regulated (*p* < 0.05), indicating that ethanol could lead to an increase in reactive oxygen species and aggravate cellular oxidative stress and lipid peroxidation. The results, as depicted in Figure 4I, show that the LJP group exhibited a significant upregulation in the expression of *acox*1, *cyp3a66*, and *fabp2* compared to the M group. This suggests that *Jiuzao* polysaccharide may ameliorate ethanol-induced liver damage by modulating genes involved in fatty acid degradation, fat digestion, absorption, and immune response. These findings are in agreement with the transcriptome analysis, thus confirming the reliability of the transcriptomic data.

### 3.4. Effect of LJP on the Intestinal Bacteria of AFLD Zebrafish Larvae

The α-diversity of the samples, as depicted in Figure 5A, revealed that the Ace index in the M group was significantly lower compared to the C, A, and LJP groups. Principal Coordinate Analysis (PCoA) was utilized to analyze differences in intestinal bacteria among the groups (Figure 5B). At the phylum level, the M group clustered to the right of the PC1 axis, while the C, LJP, and A groups clustered to the left. Similarly, at the genus level, the M group was positioned above the PC2 axis, in contrast to the C, LJP, and A groups, which were below (Figure 5C). These patterns suggest a significant alteration in the bacterial structure of the M group, while the LJP and A groups displayed more similar structures at both taxonomic levels. Redundancy Analysis (RDA) and Variance Partitioning Analysis (VPA) were conducted to assess the impact of oxidative damage on intestinal bacteria, using oxidative damage indicators as environmental variables (Figure 5D,E). RDA indicated that SOD, CAT, MDA, and GSH significantly influenced the bacterial composition of the zebrafish larvae. The VPA quantified the effects of these indicators, demonstrating that they collectively accounted for 41.77% of the variation in microbial composition, with GSH providing the highest explanatory power. However, 58.23% of the variation remained unexplained. These analyses suggest a link between oxidative damage and shifts in the gut microbiome.

Analysis of the bacterial composition at the phylum level (Figure 5(Fa)) revealed that Proteobacteria and Bacteroidota were the dominant microorganisms in zebrafish larvae. In the M group, there was a decrease in the abundance of Proteobacteria, Actinobacteriota, and Firmicutes, while the abundance of Bacteroidota increased. These changes suggest that the M group favored the proliferation of Bacteroidota while inhibiting Proteobacteria, Actinobacteriota, and Firmicutes. Conversely, the A and LJP groups promoted the growth of Proteobacteria and Firmicutes when compared to the M group and reduced the growth of Bacteroidota.

At the genus level (Figure 5(Fb)), the principal microbes identified in zebrafish larvae included *Aeromonas*, *Chryseobacterium*, *Acinetobacter*, *Pseudomonas*, *Bacillus*, and *Achromobacter*. Relative to the C group, the M group exhibited an increased abundance of Aeromonas and *Chryseobacterium*, whereas the abundance of *Acinetobacter*, *Pseudomonas*, *Bacillus*, and *Achromobacter* was decreased. These six genera might be implicated in the development of AFLD in zebrafish larvae. Furthermore, the LJP and A groups displayed a significant downregulation in the abundance of *Chryseobacterium* and *Bosea* (*p* < 0.05, Figure 5(Fc,e)) and an upregulation of *Acinetobacter* and *Pseudomonas*, compared to the M group. These findings indicate that these species are likely involved in the formation of alcoholic fatty liver and suggest that LJP may mitigate AFLD by influencing the abundance of *Chryseobacterium*, *Acinetobacter*, *Pseudomonas*, and *Bosea*.

## 4. Discussion

The liver, as the primary organ responsible for detoxification in humans, is vulnerable to damage by harmful substances, which can lead to metabolic dysfunction [28]. Excessive ethanol consumption, in particular, can result in AFLD, hepatitis, and liver cirrhosis. *Jiuzao* polysaccharides have demonstrated antioxidant capabilities [10]. The quest for benign natural antioxidants for liver disease prevention and treatment has intensified recently [29]. In our study, we extracted polysaccharides from *Jiuzao*, a winemaking by-product, and assessed their protective effects against AFLD in zebrafish larvae. Through the phenotype of the liver, it was observed that *Jiuzao* polysaccharides mitigated alcoholic hepatic steatosis in a dose-dependent manner (Figure 2). Ethanol consumption reduced the activity of SOD and CAT, increased GSH content, and elevated MDA levels in zebrafish larvae, signifying profound alterations in oxidative stress. Atomolan, known as reduced glutathione, is recognized for its hepatoprotective properties. This study revealed that both *Jiuzao* polysaccharide and atomolan could counteract the rise in MDA levels caused by alcoholic liver injury in zebrafish larvae. They also increased SOD, CAT, and GSH levels, enhancing lipid metabolism and oxidative breakdown, and improved antioxidant capacity by mitigating oxidative stress, thus contributing to hepatoprotection.

To further investigate the mitigating effects of *Jiuzao* polysaccharide on ethanol-induced AFLD in zebrafish larvae, we employed RNA-Seq technology to identify DEGs in both the C vs. M and LJP vs. M groups. The sequencing identified 2861 DEGs in the LJP group compared to the M group, with 1889 up-regulated and 972 down-regulated. GO functional annotation analysis indicated that the primary biological functions between the two groups were consistent, with approximately 3.76% of DEGs involved in immune system processes in the C and M groups and 3.60% in the LJP and M groups (Figure 4D,E).

Transcriptome analysis revealed that ethanol exposure in zebrafish larvae predominantly influenced pathways related to chemical carcinogenesis, reactive oxygen species, apoptosis, IL-17 signaling, hepatitis B, and non-alcoholic fatty liver disease. The LJP group’s gene expression alterations were chiefly in lipid metabolism and inflammatory response, differing from the M group. Notably, 33 DEGs associated with the PPAR signaling pathway, including *acox1*, *fabp2*, *cyp8b1*, *cpt1b*, *fabp1b.1*, *cpt1ab*, *acsbg1*, *ehhadh*, *acsl5*, *acadl*, *afp4*, and *apoa1b*, were up-regulated in the LJP group. Acyl-Coenzyme A oxidase 1 (*acox1*) is a downstream target of PPARα, which regulates fatty acid β-oxidation within mitochondria and peroxisomes, thus facilitating fatty acid metabolism [30]. Cytochrome p450 family 3, subfamily A, polypeptide 65 (cyp3a65) is instrumental in the oxidative stress response during ethanol metabolism and is a pivotal gene for ethanol metabolism in the zebrafish liver [31]. Fatty acid-binding protein 2 (*fabp2*) is expressed in intestinal epithelial cells, playing a role in the absorption and transport of long-chain fatty acids as well as lipid synthesis and catabolism and activating enzymes in lipid metabolism [32]. RT-qPCR confirmation of acox1 and fabp2 expression indicated their upregulation in ethanol-induced zebrafish larvae post-LJP treatment, suggesting *Jiuzao* polysaccharides chiefly regulate genes involved in lipid metabolism and oxidative stress [33]. These findings align with transcriptome results, validating their accuracy. The PPAR signaling pathway is vital for modulating the gut immune response to microbial load and diet by regulating the recruitment and activities of various immune cell populations [34]. Present data suggest ethanol-induced microbiota dysbiosis exacerbates AFLD and related metabolic disorders via inflammatory pathways [35].

Additionally, the DEGs in the LJP group participated in pathways involving fatty acid degradation, fat digestion, and absorption; xenobiotic metabolism by cytochrome P450; drug metabolism via P450; steroid biosynthesis; steroid hormone biosynthesis; and fatty acid elongation. These findings indicate that *Jiuzao* polysaccharides can potentially mitigate AFLD and related metabolic disorders by influencing organic nutrient metabolism and host inflammatory pathways [36]. The *Jiuzao* polysaccharides could reduce hepatic lipid accumulation by activating seven signaling pathways; the exact mechanisms underlying their liver-protective effects require further investigation.

Gut microbiota dysbiosis is intimately linked to metabolic diseases, and the metabolites produced by gut bacteria can impact liver injury [37,38]. It has been noted that individuals with metabolic disorders exhibit lower gut microbial diversity than healthy ones, as reflected by the significantly higher Ace index observed in the C, A, and LJP groups compared to the M group [39].

This study revealed that the M group suppressed the growth of beneficial bacteria such as Proteobacteria, Actinobacteriota, and Firmicutes while promoting Bacteroidetes. This imbalance in the intestinal bacterial ecology is associated with hepatic inflammation and lipid deposition in zebrafish larvae [40,41]. The LJP group, on the other hand, increased the abundance of Proteobacteria and Firmicutes and reduced Bacteroidetes, indicating that *Jiuzao* polysaccharides exert a corrective influence on the gut dysbiosis in zebrafish larvae with alcoholic fatty liver. Certain bacteria, such as *Pseudomonas* and *Chryseobacterium*, are known to be closely linked to inflammation and tissue infection [42]. Additionally, a reduction in Firmicutes has been correlated with high-sugar diets and lipopolysaccharide production in humans and other mammals [42,43].

In our study, we observed that treatment with *Jiuzao* polysaccharide altered the abundance of certain intestinal bacteria, namely increasing Acinetobacter and Pseudomonas and decreasing *Chryseobacterium*. These findings suggest a potential relationship between microbial changes and the balance of pro- and anti-inflammatory effects [30]. *Jiuzao* polysaccharides seem to modulate the intestinal microbiota, potentially influencing energy and lipid metabolism, as well as liver-related factors. Previous research indicates that interactions between host metabolism and gut microbiota are key in the development of fatty liver diseases [3]. Our findings suggest that *Jiuzao* polysaccharides could alleviate AFLD by affecting host gene expression and altering the intestinal microbial community. However, the exact mechanisms by which host genes and intestinal microbiota interact to impact AFLD development remain to be explained. In conclusion, *Jiuzao* polysaccharides can alleviate oxidative stress, which provides a strong basis for the development of antioxidant products using *Jiuzao* polysaccharides. In the future, the active polysaccharide extracted from *Jiuzao* is expected to be added back to *Baijiu* so as to realize high-value reuse of *Jiuzao* and provide scientific ideas for preparing healthy alcoholic beverages.

## 5. Conclusions

In this study, a zebrafish model of AFLD was developed using larvae to investigate the inhibitory effects of *Jiuzao* polysaccharides (LJP) on disease progression. It was demonstrated by the results that LJP ameliorates alcoholic hepatic steatosis in a dose-dependent manner. Transcriptomic analyses and RT-qPCR validations revealed that LJP primarily upregulates genes such as hsp90aa1.2, acox1, cyp3a66, and fabp2, which are involved in regulating fatty acid degradation, fat digestion and absorption, and immune response, thereby improving ethanol-induced liver injury. Furthermore, the alleviative effect of LJP on AFLD is attributed to its ability to modulate the intestinal microbiota, specifically by reducing the abundance of *Chryseobacterium* and *Bosea* and increasing that of *Acinetobacter* and *Pseudomonas*. These findings imply that *Jiuzao* polysaccharides might mitigate ethanol-induced fatty liver injury by influencing lipid-related gene expression and altering intestinal microbial composition. This research offers insights into the hepatic-intestinal axis in alcoholic liver injury and lays a theoretical foundation for the potential benefits of reintroducing *Jiuzao* polysaccharide into distilled *Baijiu* to reduce liver damage caused by distilled *Baijiu*.

## Figures and Tables

**Figure 1 foods-13-00276-f001:**
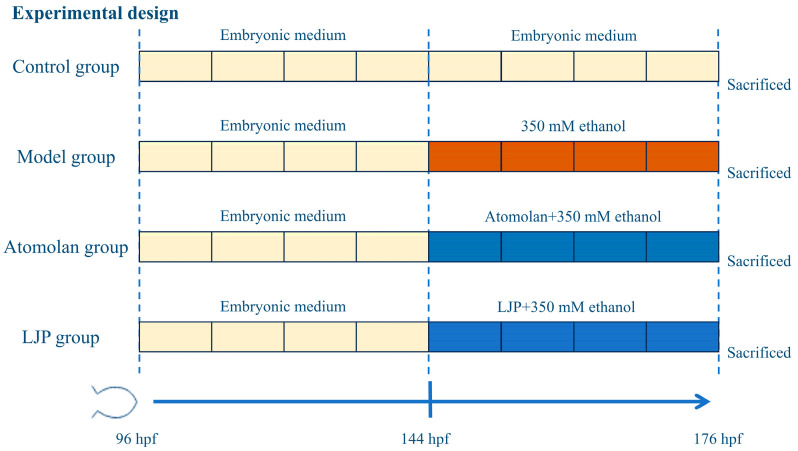
Experimental timeline for LJP feeding regime.

**Figure 2 foods-13-00276-f002:**
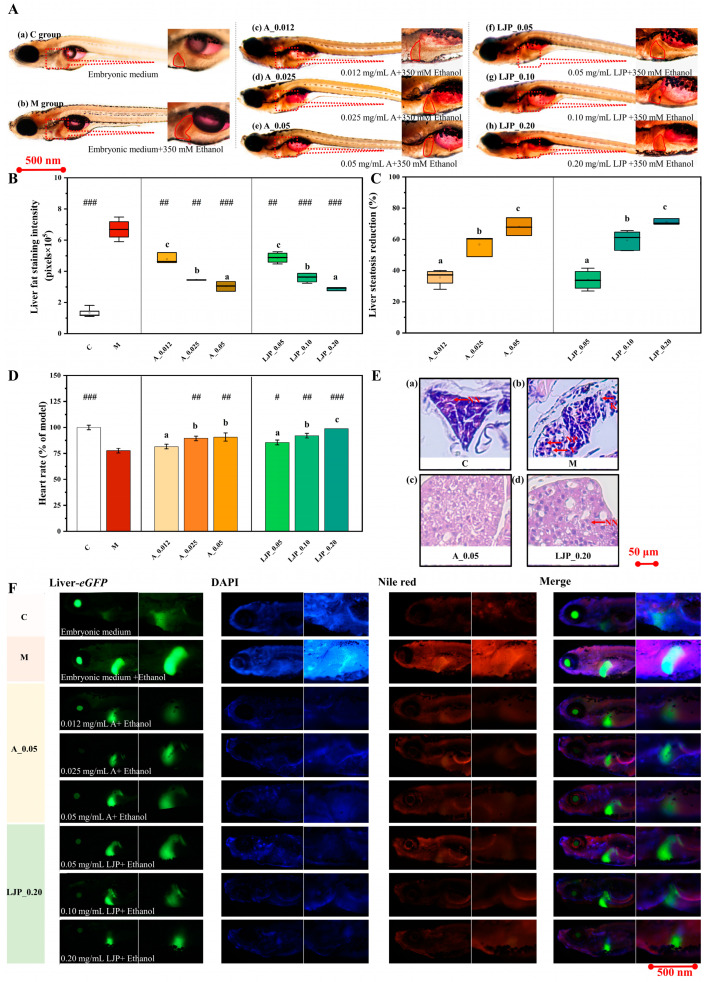
The liver visual phenotype of zebrafish larvae at 176 h post-fertilization. (**A**) Oil Red O staining was utilized to observe lipid droplets in the entire body and liver of zebrafish larvae. (**B**) Liver Oil Red O staining was quantitatively analyzed using ImageJ software (version 1.53), with results expressed as a percentage relative to the M group (mean ± SD; *n* = 3). (**C**) The inhibition rate of liver steatosis was determined by Oil Red O staining. (**D**) The impact of LJP on the cardiac rate of zebrafish larvae was evaluated. # *p* < 0.05, ## *p* < 0.01, and ### *p* < 0.001 represent significant differences compared to the M group using the unpaired 2-tailed Student’s *t*-test. Different lowercase letters indicate significant differences between groups according to the One-way ANOVA followed by post hoc Duncan’s for multiple comparisons. (**E**) Histological images of zebrafish livers from control (**a**), model (**b**), A_0.05 (**c**), and LJP_0.20 (**d**) groups. (**F**) Nile red staining, coupled with nuclear DNA staining, revealed lipid droplets in the liver cells of zebrafish larvae.

**Figure 3 foods-13-00276-f003:**
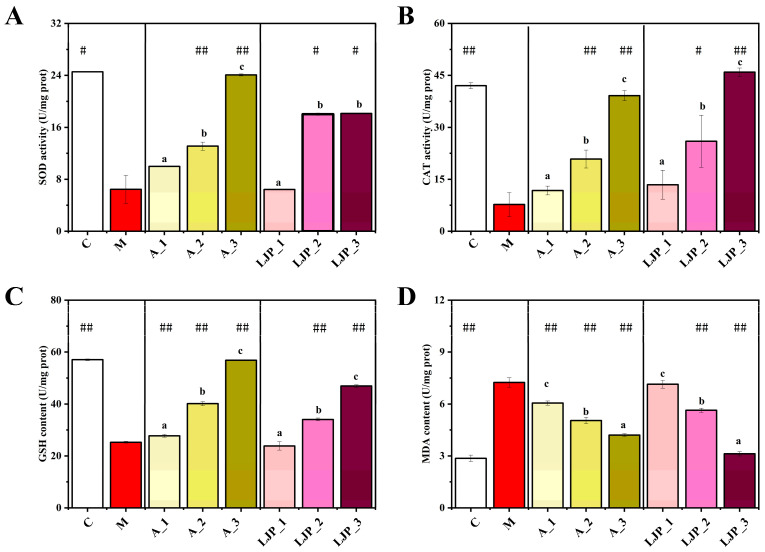
Effect of *Jiuzao* polysaccharides on the activity of enzymes related to oxidative stress in zebrafish larvae with an alcoholic fatty liver. Enzymatic activities measured included superoxide dismutase (SOD) (**A**), catalase (CAT) (**B**), glutathione (GSH) (**C**), and malondialdehyde (MDA) (**D**). The values are expressed as the mean ± SD. Statistical significance was determined using the Student’s *t*-test, where # *p* < 0.05 and ## *p* < 0.01 indicate differences compared to the M group. Distinct lowercase letters denote significant differences between the three samples according to the One-way ANOVA followed by post hoc Duncan’s for multiple comparisons.

**Figure 4 foods-13-00276-f004:**
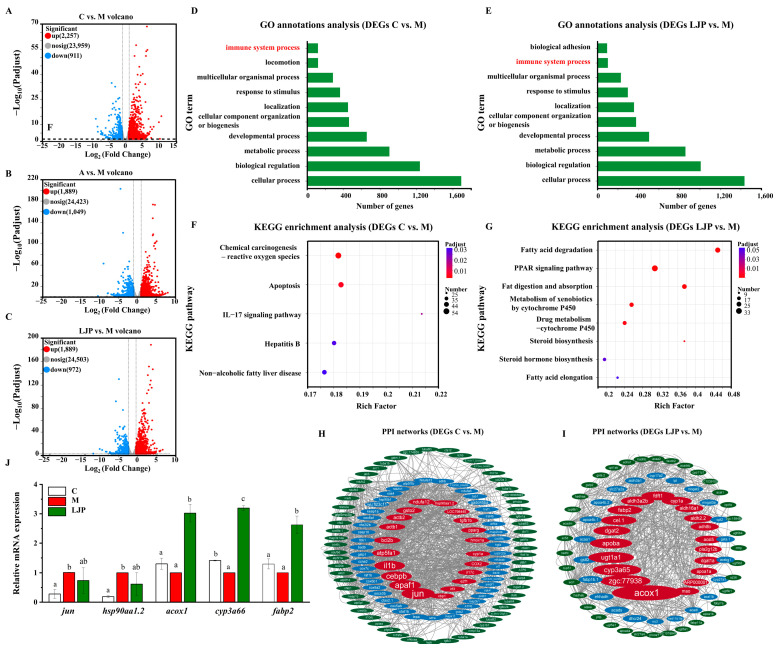
Identification of DEGs and construction of PPI network. (**A**) Volcano map of DEGs in the C vs. M group. (**B**) Volcano map of DEGs in the A vs. M group. (**C**) Volcano map of DEGs in the LJP vs. M group. (**D**) GO annotation analysis of DEGs in the C vs. M group. (**E**) GO annotation analysis of DEGs in the LJP vs. M. (**F**) KEGG enrichment analysis of DEGs in the C vs. M. (**G**) KEGG enrichment analysis of DEGs in the LJP vs. M group. (**H**) Interaction map of the protein-protein network of DEGs in the C vs. M group. (**I**) Interaction map of the protein-protein network of the DEGs in LJP vs. M group. (**J**) QRT-PCR examined the mRNA expression levels of selected genes. Different lowercase letters were significantly different between the three fractions according to the ANOVA with Duncan’s test (*p* < 0.05).

**Figure 5 foods-13-00276-f005:**
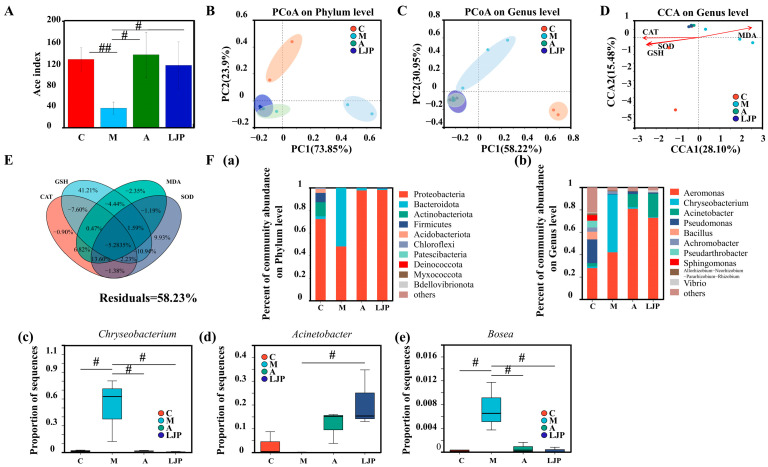
*Jiuzao* polysaccharides could improve ethanol-induced intestinal microbiota imbalance. Bar plots (**A**) illustrate the Ace index values for bacterial communities at the operational taxonomic unit (OTU) level, analyzed using Kruskal–Wallis nonparametric ANOVA, with # *p* < 0.05 and ## *p* < 0.01 indicating significance compared to the M group as per the Student’s *t*-test. β-diversity is presented through Principal Coordinate Analysis (PCoA) based on unweighted UniFrac distance metrics at the microbial phylum (**B**) and genus (**C**) levels. Canonical Correspondence Analysis (CCA) (**D**) and Variance Partitioning Analysis (VPA) (**E**) elucidate the composition and enzymatic activity correlations within the microbial community. The direction and length of the arrows in these analyses denote the strength and correlation between microbial communities and enzyme activities. The significance of differences between groups is shown in (**F**). Bar plots detail the microbial structure at both the phylum (**a**) and genus (**b**) levels. Intestinal bacteria exhibiting abundance differences at the genus level delineated in (**c**–**e**).

**Table 1 foods-13-00276-t001:** Sequences of the primers.

Target Gene	Forward Primer (5′ to 3′)	Reverse Promer (5′ to 3′)
*jun*	acacaacatgacgctcaatc	gctagactggatgatgagcc
*hsp90aa1.2*	gagagctcatctccaactcc	gctcttctttgttgggaatg
*acox1*	ctgaggctctggtggacgtg	ttgaacagtccaacaatctc
*cyp3a66*	gagaaagcttgccaaacagg	agaagcgtgtgaatcacagc
*fabp2*	catgacaacctgaagatcac	ttgtccttgcgtgtgaaagt
*rpp0*	ctgaacatctcgcccttctc	tagccgatctgcagacacac

## Data Availability

Data are contained within the article.

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
