# Peer review of "Inhibitory Effects of *Jiuzao* Polysaccharides on Alcoholic Fatty Liver Formation in Zebrafish Larvae and Their Regulatory Impact on Intestinal Microbiota"

_foods, 2024, doi:10.3390/foods13020276_

Round 1
Reviewer 1 Report
Comments and Suggestions for Authors
The polysaccharides are broadly studied currently, so the research topic is very actual, especially considering the aspect of upcycling the by-product, Jiuzao. Alcohol overconsumption and health effect of it can be considered as world-scale problem.
The abstract clearly presents the insights from the study.
line 38 reviewer suggests using "increasing the risk" instead of "heightening the risk"
line 52 "denaturation" is the process related to proteins and AA, not lipids - oxidative rancidity
line 54 how can regulate intestinal tract?
line 76 gut leaking
line 78 Gram-negative
Introduction part is quite general, can be improved. Please underline novelty of the study & practical goals.
Line 96 please supplement more info about Jiuzao.
How many larvae innicialy entered the experiment? Embryo medium was commercial or prepared on site? How the health of the larvae was evaluated?
Fig. 1 Legend is incorrect - larvae feeding? And what Authors mean by "fish water"?
Please add info about the method of larvae sacrifice.
line 189 what mean "good quality"?
line 200 why spin kit for soil, in this experiment bacteria were not environmental? Did the Authors used manure or waste water as sample?
How the gut biota was collected?
post hoc statistical analysis in groups?
The M&M part needs corrections and completing
The statistical markings on Figs & idea of analysis are not clear for the reviewer
Paragraph 3.2 - tests were not included into M&M part
Genomic analysis results& bacteria profiling are well presented.
lines 406-409??? can be removed
line 420 which markers of apoptosis?
Fig. 2 panel A is not showing well important details, should be improved.
Discussion must be improved - how this study findings reflecting current state of knowledge in the field, give more interpretations of the results, future perspectives.
Conclusions are based on study findings, just please avoid too far reaching statements about prevention &treatment of the disease
Comments on the Quality of English Languagegrammar is fine, please check terminology used in the manuscript
Reviewer 2 Report
Comments and Suggestions for Authors
The manuscript entitled, "Inhibitory Effects of Jiuzao Polysaccharides on Alcoholic Fatty Liver Formation in Zebrafish Larvae and Their Regulatory Impact on Intestinal Microbiota" is innovative study, however the experimental design can be improved by considering following suggestions.
In experimental design zebrafish larvae were exposed to ethanol and polysaccharides to observe the influence on fatty liver induced by alcohol. However the model used is not very appropriate for the claim made here. Why authors specifically used larval stage?? why not mature fish?
How from the figure it can be observed that there was significant reduction in liver fat after treatment with LJP.
Morphological prediction alone may not support the findings here, what was the mechanism of action authors are proposing to justify this pharmacological response??
Rewrite the conclusion, rather than summarizing the study like abstract, mention the concrete findings of the study.
Comments on the Quality of English Language
Minor language corrections are needed throughout the document.
Reviewer 3 Report
Comments and Suggestions for Authors
Inhibitory Effects of Jiuzao Polysaccharides on Alcoholic Fatty Liver Formation in Zebrafish Larvae and Their Regulatory Impact on Intestinal Microbiota
Qing Li et at
The liver is critical in alcohol metabolism, and excessive consumption heightens the risk of hepatic damage, potentially escalating to hepatitis and cirrhosis. Jiuzao, a by-product of Baijiu production, contains a rich concentration of naturally active polysaccharides known for their antioxidative properties. This study investigated the influence of Laowuzeng Jiuzao Polysaccharide (LJP) on the development of ethanol-induced alcoholic fatty liver. Zebrafish larvae served as the model organisms for examining the LJP's hepatic impact via liver phenotypic and biochemical assays. Additionally, the study evaluated the LJP's effects on gene expression associated with alcoholic fatty liver and the composition of the intestinal microbiota through transcriptomic analysis and 16S rRNA sequencing, respectively. Our findings revealed that LJP markedly mitigated morphological liver damage and reduced oxidative stress and lipid peroxidation in larvae. Transcriptome data indicated that LJP ameliorated hepatic fat accumulation and liver injury by enhancing gene expression involved in alcohol and lipid metabolism. Furthermore, LJP modulated the development of alcoholic fatty liver by altering the prevalence of intestinal Acinetobacter and Firmicutes, specifically augmenting Acinetobacter while diminishing Chryseobacterium levels. Ultimately, LJP mitigated alcohol-induced hepatic injury by modulating gene expression related to ethanol metabolism, lipid metabolism, and inflammation and by orchestrating alterations in the intestinal microbiota.
Comment to authors:
The paper has a standard format with meticulous experimental design. This study investigated the impact of Laowuzeng Jiuzao Polysaccharide (LJP) on the development of ethanol-induced alcoholic fatty liver. Moreover, Ethanol-induced alcoholic fatty liver is becoming more prevalent worldwide. I believe that the information presented in the paper will captivate readers and generate numerous citations. Therefore, I am very interested and impressed with this manuscript. The English in the manuscript is also of high quality.
Round 2
Reviewer 2 Report
Comments and Suggestions for Authors
The manuscript is revised as advised.